# Isolation of Peptide Inhibiting SGC-7901 Cell Proliferation from *Aspongopus chinensis* Dallas

**DOI:** 10.3390/ijms232012535

**Published:** 2022-10-19

**Authors:** Xu-Mei Chen, Shu-Qi Zhang, Mi-Lan Cao, Jian-Jun Guo, Rui Luo

**Affiliations:** 1Scientific Observing and Experimental Station of Crop Pests in Guiyang, Ministry of Agricultural and Rural Affairs, Institute of Entomology, Guizhou University, Guiyang 550025, China; 2College of Life Science, Guizhou University, Guiyang 550025, China

**Keywords:** anti-cancer peptide, *Aspongopus chinensis* Dallas, MTT, inhibition, IC_50_

## Abstract

*Aspongopus chinensis* Dallas is used as a traditional Chinese medicine as well as an edible insect. Although its anti-tumor effects have been observed, the anti-tumor active component(s) in the hemolymph of *A. chinensis* remains unknown. In this study, a combination usage of ultrafiltration, gel filtration chromatography, FPLC and RP-HPLC to separate and purify active peptides was performed based on the proliferation of the human gastric cancer SGC-7901 cell line treated with candidates. One peptide (MW = 2853.3 Da) was isolated from the hemolymph of *A. chinensis.* A total of 24 amino acid residues were continuously determined for the active peptide: N′-ECGYCAEKGIRCDDIHCCTGLKKK-C′. In conclusion, a peptide that can inhibit the proliferation of gastric cancer SGC-7901 cells in the hemolymph of *A. chinensis* was purified in this study, which is homologous to members of the spider toxin protein family. These results should facilitate further works for this peptide, such as the cloning of genes, expression in vitro by prokaryotic or eukaryotic systems, more specific tests of anti-tumor activity, and so on.

## 1. Introduction

Cancers are a serious threat to human health. There were 19.3 million new cancer cases and 10 million cancer deaths in 2020 [1]. Lung cancer, colorectal cancer, breast cancer, gastric cancer, and liver cancer are the top five cancers with the highest morbidity and mortality worldwide [2]. However, there are no adequate efficacious medicines for cancer treatments. Meanwhile, many disadvantages are associated with current medicines, such as high price, severe adverse reactions, drug resistance, relapse after drug cessation, and so on [3]. Therefore, the development of new anti-cancer drugs is a major issue at present.

Development of anti-cancer chemicals should be based on chemical design and screening according to targets, or based on the screening of natural chemicals. For natural anti-cancer chemical screening, medicinal insects are important resources for active component isolation. It has been proven that cantharidin and its derivatives from meloid insects possess inhibitory effects on various cancer cells, such as laryngeal cancer, gastric cancer, leukemia, esophageal cancer, liver cancer, lung cancer, cervical carcinoma and prostatic cancer [4,5,6,7,8,9]. Bee venom and propolis (Apidae and Vespidae, Hymenoptera), which are traditionally utilized as medicines, also have inhibitory effects on the proliferation of leukemia, liver cancer, and esophageal carcinoma cells [10,11]. Mastoparan is an α-helical and amphipathic tetradecapeptide obtained from the venom of the wasp *Vespula lewisii* that exhibits tumor cell cytotoxicity [12]. It induced caspase-dependent apoptosis in melanoma cells through the intrinsic mitochondrial pathway, protecting mice against tumor development. In addition, chemicals originating from *Bombyx mori*, *Chrysomya megacephala*, *Musca domestica*, and *Holotrichia diomphalia* larvae also have certain inhibitory effects on various tumor cells [13,14,15,16].

Anti-cancer peptides are novel natural anti-cancer chemicals [17,18]. Insects are important sources of anti-cancer peptides, such as mastoparan mentioned above from *V. lewisii* [19]; Mastoparan-C from *Vespa crabro*, which can inhibit the proliferation of non-small cell lung cancer H157, melanocyte MDA-MB-435S, human prostate carcinoma PC-3, human glioblastoma astrocytoma U251MG, and human breast cancer MCF-7 cells [20]; SK84 from *Drosophila virilis* larvae, with a specific inhibitory effect on the proliferation of various cancer cell lines such as human leukemia THP-1, liver cancer HepG2, and breast cancer MCF-7 cells [21]. Contents of biological active peptides in organisms are usually very low [22]. Moreover, these peptides are usually mixed with sophisticated components in various tissues. Common techniques to purify these peptides include gel filtration chromatography, ion exchange chromatography, reversed-phase high performance liquid chromatography (RP-HPLC), affinity chromatography, and so on [23].

*Aspongopus chinensis* Dallas 1851 is a Traditional Chinese Medicine and also acts as edible material in China. It was found that the hemolymph of *A. chinensis* could significantly inhibit the growth of human gastric cancer cell SGC-7901 and human breast cancer cells in a time- and dose-dependent manner [24,25,26,27,28]. To ascertain the active protein/peptide component(s) in hemolymph of *A. chinensis* for cancer resistance, ultrafiltration, gel filtration chromatography, ion exchange chromatography, and RP-HPLC were sequentially used in this study. Inhibition of cell proliferation (human gastric cancer cells SGC-7901) was tested for sections isolated at each stage. The anti-tumor peptide isolated from *A. chinensis* is named as AcATP (anti-tumor peptide of *A. chinensis*). Furthermore, the molecular weight of AcATP was determined by Matrix-Assisted Laser Desorption/Ionization-Time of Flight Mass Spectrometry (MALDI-TOF-MS), and its amino acid sequence was determined based on the Edman chemical degradation method. The results of this study should facilitate research on the cloning of active peptides, biological active tests, recombinant protein production, and so on.

## 2. Results

### 2.1. Inhibiting Effect of Ultrafiltration Components in Hemolymph of A. chinensis on SGC-7901 Cells

The hemolymph of *A. chinensis* was divided into three sections: Sect. I (molecular weights < 3 kD), Sect. II (3–50 kD), and Sect. III (>50 kD). The proliferation inhibiting effect of each section on SGC-7901 cells was observed by an inverted microscope. Except for reductions of cell density, cultured cells that were treated with separated components also showed other abnormal phenomena: shrinkage, smaller volume, nucleus pyknosis, shedding, and floating (Figure 1a). Cytotoxic effects to SGC-7901 cells resulting from these three sections (treatments with 100, 200, 400 μg/mL, respectively) are shown in Table 1. For each section, the inhibition on cell proliferation became more effective with an increase of each section supplement. At the same concentration, treatment with Sect. III resulted in less inhibition of cell proliferation (significantly or non-significantly). At a concentration of 400 μg/mL, Sect. II had the highest inhibition rate (56.85 ± 3.79%) on SGC-7901 cells; the IC_50_ (50% inhibitory concentration) of Sect. II was 358.600 μg/mL (Figure 1b). Furthermore, cell morphological changes treated with Sect. I or II were more obvious than Sect. III. Therefore, Sect. II was chosen to separate active peptides in the following steps.

### 2.2. Inhibiting Effect of Components Separated by Gel Filtration Chromatography

Five fractions in Sect. II (3–50 kD, ultrafiltration) were discriminated and eluted (Figure 2a): peaks 1–5 were denoted as Sect. II-1–5, respectively. Each fraction (differential peak collection) was desalted, freeze-dried, and re-dissolved to various concentrations to test the inhibition effects on SGC-7901 cells. Except for Sect. II-5 (peak 5) at lower concentration (25 μg/mL), the fractions expressed inhibitory effects (Figure 2b). Although Sect. II-3 (peak 3) and Sect. II-2 (peak 2) both expressed stronger inhibition, there was not further enhancement of the inhibition effect along with increasing dosage for Sect. II-2. Sect. II-3 had more effective inhibition on proliferation of SGC-7901 cells (with a dosage-dependent manner) than other fractions, especially at high concentrations (100 μg/mL), and the IC_50_ value was 155.200 μg/mL (Figure 2c). Therefore, Sect. II-3 was used for further separation or purification.

### 2.3. Inhibition Effect of Components Separated by FPLC

The peptides/proteins bound to the Resource Q strong anion exchange resin (ÄKTA Purifier FPLC system) were linearly eluted using 0–1 M NaCl elution buffer for 50 min. Four fractions in Sect. II-3 (gel filtration chromatography) were discriminated and eluted (Figure 3a): peaks 1–4 were denoted as Sects. II-3-1-4, respectively. Each fraction (differential peak collection) was treated the same as above. Except for Sect. II-3-4 (peak 4) at a lower concentration (10 μg/mL), the fractions expressed inhibitory effects (Figure 3b). It was found that Sect. II-3-3. (peak 3) was more effective than the other fractions at any concentration, and its IC_50_ value for SGC-7901 cells was 70.900 μg/mL (Figure 3c). Therefore, Sect. II-3-3 was collected for further separation or purification.

### 2.4. Inhibition Effect of Components Separated by RP-HPLC

Six fractions in Sect. II-3-3 (FPLC) were discriminated and eluted by RP-HPLC (Figure 4a): peaks 1–6 were denoted as Sect. II-3-3-1-6, respectively. Each fraction (differential peak collection) was treated the same as above. Sect. II-3-3-2 (peak 2), Sect. II-3-3-3 (peak 3), and Sect. II-3-3-5 (peak 5) expressed inhibitory effects on the proliferation of SGC-7901. In particular, a dosage-dependent action manner was shown in Sect. II-3-3-2 and Sect. II-3-3-5 (Figure 4b). Sect. II-3-3-5 had the highest inhibition effect at a high concentration treatment (30 μg/mL), and its IC_50_ value was 51.290 μg/mL (Figure 4c). Therefore, Sect. II-3-3-5 was collected to determine molecular weight and amino acid sequence.

Although the raw product (Sect. II, isolated by ultrafiltration) originating from the hemolymph of *A. chinensis* had a relatively high quantity, the component(s) with anti-cancer activity was had a low yield rate (Table 2). The final yield rate of active peptide(s) was 0.020% compared to peptides/proteins in the raw product (Sect. II). Compared to the last step, the yield rate for each dissection operation increased along with separation and purification, step by step. Along with the purification and enrichment of active peptide(s), the IC_50_ value of isolated fractions decreased sequentially.

### 2.5. Molecular Weight and Amino Acid Sequence of Active Peptide

Molecular weight of the active peptide (Sect. II-3-3-5) that inhibited proliferation of gastric cancer SGC-7901 cells was determined as 2853.3 Da by MALDI-TOF-MS (Figure 5a). Amino acid sequence of the active peptide was determined by the ‘Edman chemical degradation method’. Together, 24 amino acid residues were continuously determined for the active peptide (Figure 5b, Appendix A): N′-EKHYAAEKGIRRDDIIHTTTLKKK-C′.

## 3. Discussion

Cancer is the second leading cause of death after cardiovascular and cerebrovascular diseases and poses serious threats to human life and health. Therefore, scholars at home and abroad have been working to explore ways to treat cancer. In recent years, due to the continuous research on anti-cancer traditional Chinese medicine, the advantages of the comprehensive conditioning function, lower toxicity, and fewer side effects of anti-cancer medicinal insects have attracted more and more attention. At present, several drugs have been developed and used in the treatment of cancer. Medicinal insects have a significant inhibitory effect on a variety of cancer cells, with different types of anti-cancer mechanisms, such as signaling pathways regulating apoptosis of tumor cells, blocking of the tumor cell cycle, and inhibition of the migration and invasion of tumor cells. The diversity of anticancer mechanisms not only facilitates the development and utilization of medicinal insects from multiple angles but also reduces the possibility of drug resistance in tumor cells. However, there are still many problems to be solved to realize the widespread application of anti-cancer insects. Meanwhile, the components of anti-cancer insects are complex, and their anti-cancer functions and mechanisms are not yet fully understood.

*A. chinensis* is an anti-cancer medicinal insect. Previous experiments have shown its significant anti-cancer activity of inhibiting proliferation of human gastric cancer SGC-7901 cells and breast cancer MCF-7 cells [27]. While, its anti-cancer active component(s) is unclear. In this study, the combination of ultrafiltration, gel filtration chromatography, FPLC, and RP-HPLC is utilized to separate and purify the active peptide based on proliferation of human gastric cancer SGC-7901 cells. The active peptide possesses a molecular weight of 2853.3 Da. As seen in the results in this study, the active peptide from *A. chinensis* inhibits cell proliferation of SGC-7901 cells. It is suggested that this peptide is one of the active substances of *A. chinensis* and may participate in the anticancer effect of the hemolymph of the *A. chinensis.* The results of this study also provide an experimental basis for the development of anti-tumor substances with hemolymph activity of the nine incense worms, and provide a new idea for the development of anti-tumor aversion of the nine incense worms.

The hemolymph is more effective for inhibition of SGC-7901 cells than other tissues of *A. chinensis*. The protein/peptide section is more effective for inhibition of SGC-7901 cells than other constituents in the hemolymph [29]. However, the content of the protein/peptide in the hemolymph of *A. chinensis* is very low. Separation and purification of the peptide (antitumor active component) in this study are valid, which is indicated by the decreasing of IC_50_ values for sequential separating products (Table 2). The final yield rate of the peptide (compared to 3–50 kD product by ultrafiltration) is very low (0.020%). The very low yield rate of active components for each separation/purification operation is not consistent with the change of IC_50_ values for the respective product (Table 2). This result could be as a result of the partial recovery of active components in each operation or from losses of active components that possess minor or auxiliary active components in each separation/purification step (Figure 3a, Figure 4a and Figure 5a,b).

In summary, this study was the first to isolate and purify an active peptide that inhibits the proliferation of gastric cancer cells from the hemolymph of the worm, which lays an important foundation for the development of the medicinal insect resources and the research and development of new anti-tumor drugs.

## 4. Materials and Methods

### 4.1. Materials

*A. chinensis* was collected from Xiasi Town, Kaili City, Guizhou Province, China (107°80′ E, 26°52′ N), and stored in −80 °C refrigerators. Their bodies were placed into a 10 mL syringe after removing heads, chests, and wings and then squeezed to release the hemolymph. The hemolymph was transferred into centrifuge tubes. After centrifugation at 12,000× *g* for 30 min at 4 °C, the top liquid layer (containing oil) was gently scraped off. The aqueous solution was centrifuged again at 12,000× *g* for 30 min at 4 °C, and the supernatant was collected for further research.

### 4.2. Protein Concentration Determination by BCA

Measurements of protein concentration for the original hemolymph of *A. chinensis* and separated products in were carried out using the BCA method and the BCA protein concentration assay kit instructions (Cat. No.: BL521A, Biosharp). A standard curve (absorbance value at 562 nm) for determination of protein concentrations was made with a series of bovine serum solutions dissolved with PBS.

### 4.3. Cell Culture and MTT Assay

Human gastric cancer SGC-7901 cells (Shanghai Cell Bank of Chinese Academy of Sciences) were cultured in a DMEM (high glucose) medium containing 10% fetal bovine serum (FCS) and 1% streptomycin in a carbon dioxide incubator (37 °C, 5% CO_2_). The medium was renewed every other day during culture, and sub-cultures of cells were carried out when cells were spread to 80% on the bottom of flask (in the logarithmic growth phase).

MTT [3-(4,5-dimethyl thiazol-2-yl)-2,5-diphenyl tetrazolium bromide] assay was carried out to detect the inhibition effect resulting from the hemolymph of *A. chinensis* from Section 2.1 or other dissected products that were added into the cultural medium after sub-cultured cells, adherent with four replicates per group. Inhibition rate = [1 − (P1-O)/(P0-O)] × 100%. P1 is a value of the experiment group (SGC-7901 cells treated with eluted fraction); P0 is a value of the cell group without the drug (SGC-7901 cells without treatment); O is a value of the blank group (medium without cells or drug, only cell culture fluid). The experiment was repeated three times. The half-inhibitory concentration (IC_50_ value) was calculated by curve fitting.

### 4.4. Separation and Purification of Active Component in Hemolymph of A. chinensis

#### 4.4.1. Ultrafiltration

The hemolymph of *A. chinensis* was diluted with PBS at a ratio of 1:4 and then centrifuged for 1 h (12,000× *g*, 4 °C) to collect the supernatant. The supernatant was transferred to MWCO 50 kD and MWCO 3 kD ultrafiltration tubes (Millipore) and centrifuged for 1–2 h (4500× *g*, 4 °C). Components in the hemolymph were separated into three sections: >50 kD, 3–50 kD, and <3 kD. After freeze-drying, these components were redissolved in 1 mL PBS.

#### 4.4.2. Gel Filtration Chromatography

The Sephadex G-50 cross-linked dextran gel was selected as the filler (GE Healthcare of American) to separate components in the product using ultrafiltration. The equilibration/elution buffer was 50 mmol/L Tris-HCl (pH 9.0). The OD_280_/OD_215_ was measured with a spectrophotometer (Ultrospec 2100 pro UV/Vis, Amersham Bioscience; the same as below) along with elution. Eluted samples were collected and combined according to the peak shape and then lyophilized. After freeze-drying, collections for each peak component were redissolved in 1 mL PBS.

#### 4.4.3. Fast Protein Liquid Chromatography (FPLC)

To separate components in the product that resulted from gel filtration chromatography, FPLC (ÄKTA Explorer 10S Purifier FPLC, GE) was used. The following were used in the experiment: chromatography column: resource Q (6 mL) strong anion exchange column; equilibration buffer (solution A): 50 mmol/L Tris-HCl (pH 9.0); elution buffer (solution B): 50 mmol/L Tris-HCl + 1 mol/L NaCl (pH 9.0); pressure limit: 0.6 MPa. A linear gradient elution was performed with 0–1 mol/L NaCl elution buffer at a flow rate of 1.5 mL/min for 50 min. The OD_215_/OD_280_ was measured along with elution. Eluted samples were collected and combined according to the peak shape and then lyophilized. Further treatments for collections are the same as above.

#### 4.4.4. Reverse-Phase High Performance Liquid Chromatography (RP-HPLC)

To separate components in the product that resulted from FPLP, RP-HPLC (Waters 1525 series Binary HPLC Pump, Waters) was used. The following were used: preparation column: Waters X BridgeTM C18 Size LC Column (4.6 × 250 mm); equilibration buffer (solution A): 99.9% ultrapure water + 0.1% trifluoroacetic acid (TFA); elution buffer (solution B): 99.9% acetonitrile + 0.1% trifluoroacetic acid (TFA). The procedure for gradient elution is shown in Table 3. The OD_215_/OD_280_ was measured along with elution. The UV absorbance values of OD_215_ and OD_280_ were measured using a Waters 2489 Vis/UV detector. Eluted samples were collected and combined according to the peak shape and then lyophilized. Further treatments for collections are the same as above.

### 4.5. Determination of Molecular Weight and Amino Acid Sequencing for Active Peptide in Hemolymph of A. chinensis

#### 4.5.1. Determination of Molecular Weight for Peptide(s)

The desalting active peptide(s) (10 µL, >4 pmol/µL) resulting from RP-HPLC was used to detect the purity and then was used for molecular weight measurement by matrix-assisted laser desorption time-of-flight mass spectrometry [30].

#### 4.5.2. Amino Acid Sequencing

The pure active peptide(s) (about 100 µg) resulting from RP-HPLC was used to determine its amino acid sequence with the Edman chemical degradation method [19].

### 4.6. Data Analysis

Data were analyzed by one-way ANOVA analysis and homogeneity testing of variance using statistical software package SPSS 19.0. The variance t-test was used to compare the data of multiple groups. All data were expressed by $\bar{X}$ ± SD (Standard Deviation), and the difference was statistically significant at *p* < 0.05.

## 5. Conclusions

In conclusion, a peptide could inhibit the proliferation of gastric cancer SGC-7901 cells in the hemolymph of *A. chinensis*, separated and purified by ultrafiltration, gel filtration chromatography, FPLC, and RP-HPLC in this study. We determined that the sequence of the active peptide is N′-EKHYAAEKGIRRDDIIHTTTLKK-C′, and its molecular weight is 2853.3 Da. These results should facilitate further works for this peptide, such as the cloning of genes, expression in vitro by prokaryotic or eukaryotic systems, additional testing of anti-tumor activity, and so on.

## Figures and Tables

**Figure 1 ijms-23-12535-f001:**
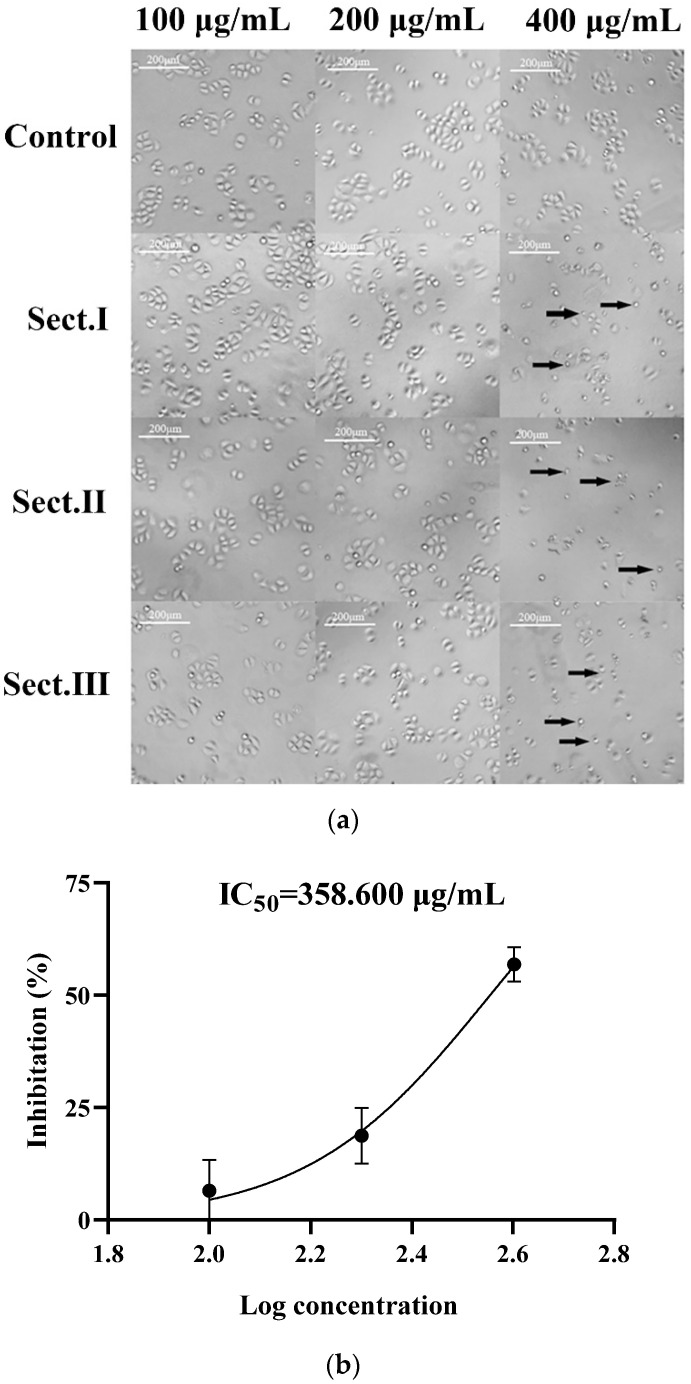
(**a**) Effects of ultra-filtration components in hemolymph of *A. chinensis* on SGC-7901 cells after 48 h treatment. Control: cell group without drug; Sect. I: molecular weights of the section < 3 kD; Sect. II: molecular weights of the section between 3 kD and 50 kD; Sect. III: molecular weights of the section > 50 Kd, the cells indicated by the arrow are morphologically altered cells. (**b**) IC_50_ curve fitting of Sect. II.

**Figure 2 ijms-23-12535-f002:**
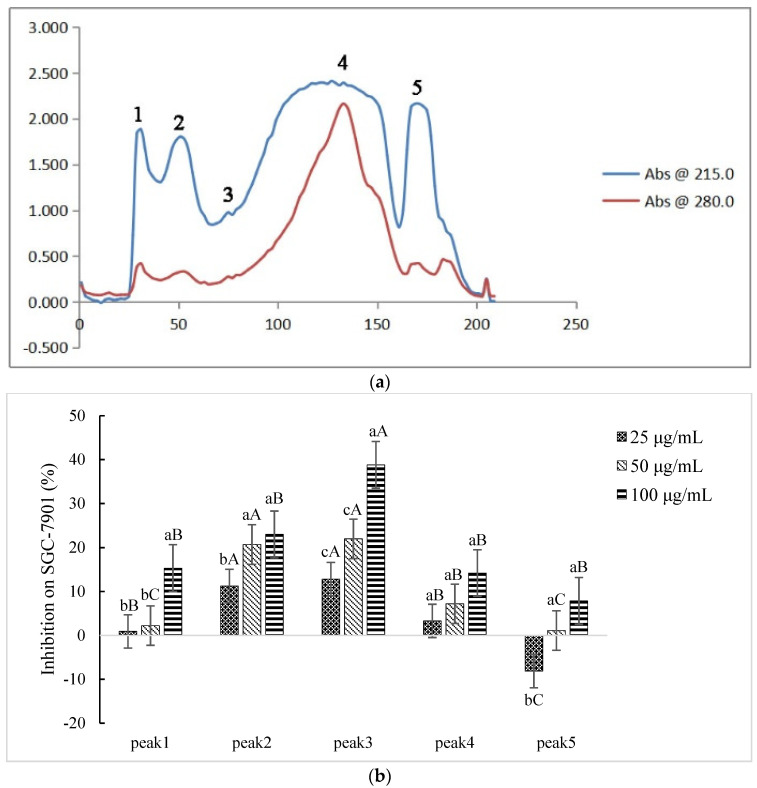
Inhibiting effect of components separated by gel filtration chromatography. (**a**) The elution curve of gel filtration chromatography. (**b**) Inhibitory rates of cell proliferation (SGC-7901 cells) induced by Sect. II-1–5 that were separated by gel filtration chromatography after 48 h treatment. a, b, c means *p* < 0.05; A, B, C means *p* < 0.01. (**c**) IC_50_ curve fitting of Sect. II-3.

**Figure 3 ijms-23-12535-f003:**
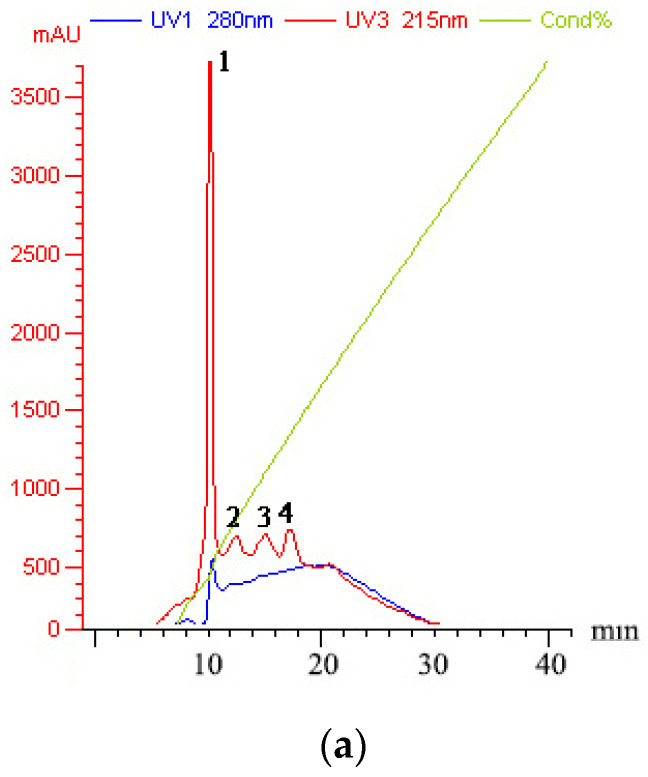
Inhibition effect of components separated by FPLC. (**a**) The elution curve of FPLC, linear electrical conductivity showed correctly linear elution. (**b**) Inhibitory rates of cell proliferation (SGC-7901 cells) induced by Sect. II-3-1-4 that were separated by FPLC after 48 h treatment. a, b, c means *p* < 0.05; A, B, C means *p* < 0.01. (**c**) IC_50_ curve fitting of Sect. II-3-3.

**Figure 4 ijms-23-12535-f004:**
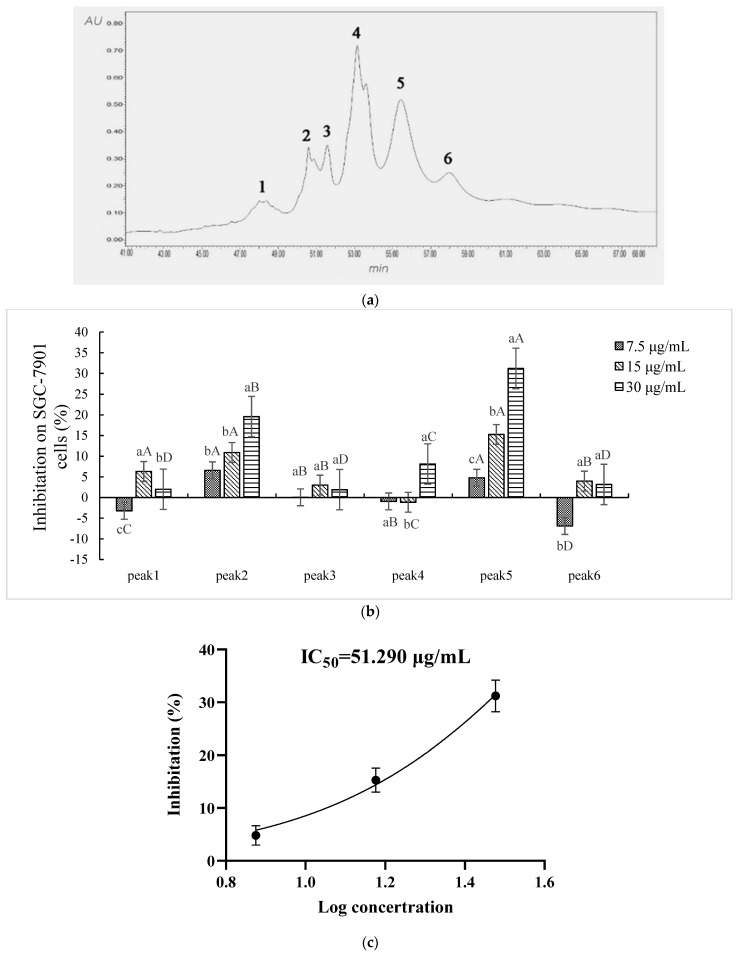
Inhibition effect of components separated by RP-HPLC. (**a**) The elution curve of RP-HPLC. (**b**) Inhibitory rates of cell proliferation (SGC-7901 cells) induced by Sect. II-3-3-1~6 that were separated by RP-HPLC after 48 h treatment. a, b, c means *p* < 0.05; A, B, C, D means *p* < 0.01. (**c**) IC_50_ curve fitting of Sect. II-3-3-5.

**Figure 5 ijms-23-12535-f005:**
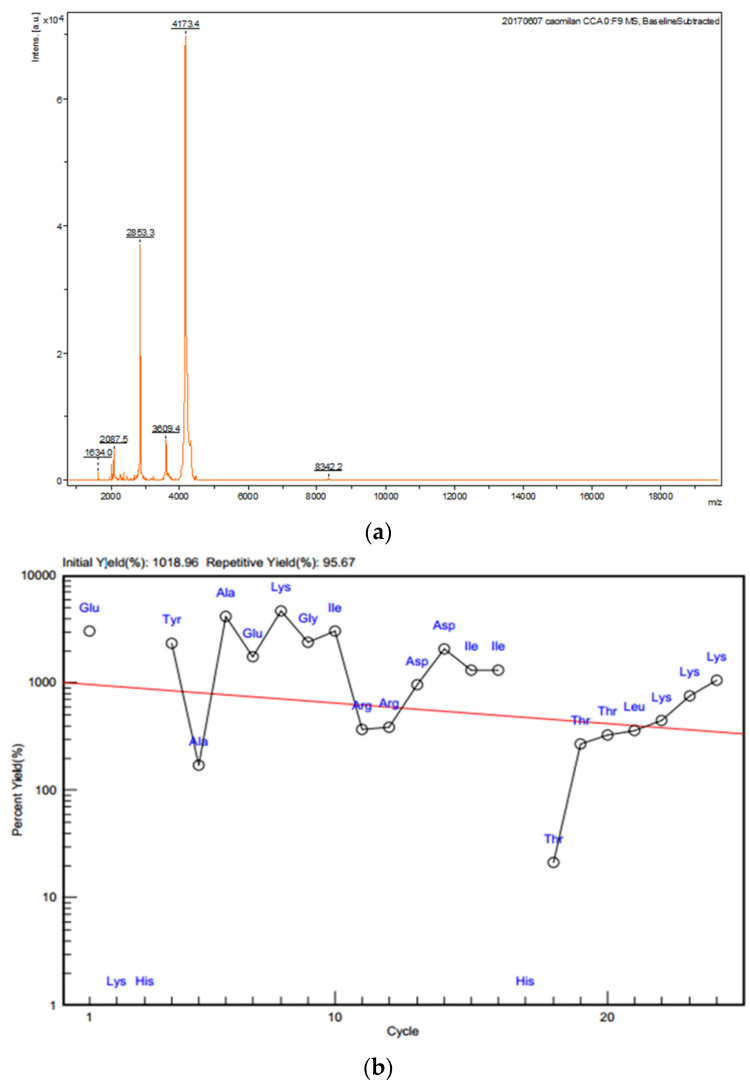
Molecular weight and amino acid sequence of active peptide. (**a**) Molecular weight of the active peptide determined by MALDI-TOF-MS. (**b**) *N*-terminal amino acid sequence of active protein.

**Table 1 ijms-23-12535-t001:** Inhibitory rates of cell proliferation (SGC-7901 cells) induced by various components of *A. chinensis* hemolymph separated by ultrafiltration after 48 h treatment.

Sample Concentration (μg/mL)	Inhibitory Rate (%)		
	Sect. I (<3 kD)	Sect. II (3~50 kD)	Sect. III (>50 kD)
100	16.13 ± 6.04 bA	6.45 ± 6.91 cB	−0.10 ± 5.11 cC
200	20.90 ± 15.04 bA	18.72 ± 6.16 bA	11.71 ± 4.60 bA
400	46.77 ± 5.79 aA	56.85 ± 3.79 aA	32.89 ± 4.09 aB

Test was repeated four times (n = 4). a, b, c means *p* < 0.05; A, B, C means *p* < 0.01.

**Table 2 ijms-23-12535-t002:** Effectiveness of various separation operations denoted by protein yield and IC_50._

Purification Step	Total Protein Quantity (mg)	Yield Rates Compared to Ultrafiltration (%)	Yield Rates Compared to Former Step (%)	IC_50_ (μg/mL)
Ultrafiltration	1753.500	-	-	358.600
Gel filtration chromatography	26.736	1.524	1.524	155.200
FPLC	2.307	0.132	8.629	70.900
RP-HPLC	0.359	0.020	15.561	51.290

**Table 3 ijms-23-12535-t003:** Elution procedure of RP-HPLC.

Time (min)	Buffer A (%)	Buffer B (%)	Flow Rate (mL/min)
0–5	95	5	0.7
6–115	40	60	0.7
116–125	0	100	0.7
126–130	95	5	0.7

## Data Availability

All the data are available within the manuscript.

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
