# Peer review of "Isolation of Peptide Inhibiting SGC-7901 Cell Proliferation from Aspongopus chinensis Dallas"

_ijms, 2022, doi:10.3390/ijms232012535_

Round 1

Reviewer 1 Report

Please see attached comment

Reviewer 2 Report

Chen and coauthors report the isolation of anticancer peptide active component from medicinal insect Aspongopus chinensis Dallas using human gastric cancer cell line SGC-7901. The overall studies are scientifically sound including the tandem purification procedures to enrich the active peptide but a few concerns are listed below for consideration.

(1) Figure 5b, it's unclear how the figure can support the determined peptide sequence of interest: N'-ECGYCAEKGIRCDDIHCCTGLKKK-C'. Some detailed technical explanation is needed through careful examination.

(2) Page 9, "The half-inhibitory concentration (IC50 value) was calcu- 258 lated by SPSS 19.0. lgIC50 = Xm-I [P-3(P-Pm-Pn)/4]. Xm is lg maximum dose; I is lg (max- 259 imum dose/adjacent metering); P is the sum of positive reaction rates Pm is the maximum 260 positive reaction rate; Pn is the minimum positive reaction rate."

It's recommended that the IC50 shall be calculated based a full dose response curve with at least 6 concentrations and can be subject to a more common way of curve fitting.

(3) A control human cell line (such as HEK) shall be used to access the cell inhibition effect and in vitro toxicity of the isolated active peptide to compare with SGC-7901 that's cultured under the same condition and experimented with the same dose response manner as SGC-7901.

(4) Once the first and second concerns above are addressed, the authors may also include nM IC50 besides ug/ml. 

Round 2

Reviewer 1 Report

Please see attached comments on the authors' answers.

Reviewer 2 Report

The authors provided additional information on Edman degradation sequencing results but certain amino acid residuals identity appears not reliable based on the sequencing reports. Either a solid explanation how these sequences are derived based on the sequencing results or it should be mentioned in the text about any uncertainty about particular amino acid residual positions as a clarification.

The other suggestion is that if IC50 is indeed derived from dose response curve, such curve fitting shall be shown in the main text figures. 

Round 3

Reviewer 1 Report

The authors made significant alterations and corrections to the manuscript, making the text more attractive and impactful. Therefore, I believe that this new version of the manuscript is suitable for publication. Therefore, my opinion is in favor of publishing the manuscript in this last version.

Author Response

 Thank you for your comment.

Reviewer 2 Report

The authors attempted to address the concerns but new concerns are raised. The work itself is still interesting and may have potential impact in the field, but the conclusions need additional support.

(1) "We have determined that the sequence of the active peptide is N '- EKHYAAEKGIRRDDIIHTTTLKK-C', and its molecular weight is 4177.6 Da. "

This conclusion has a problem. The claimed sequence of the active peptide (calculated to be  2724.12 Da based on the claimed sequence) is not consistent with the molecular weight 4177.6 Da). I suggest either trouble shoot Edman degradation to ensure reliability of the amino acid residues' identities or using some kind of tandem mass spec to figure out the exact peptide sequence or the molecule's identity, or perhaps it's a dimer of some sort after correct sequence is assigned.

(2) In order to be considered generally acceptable in the field of cell inhibition studies, IC50 needs to be determined from dose response curve of at least 5-6 concentrations to capture both the IC50 range and the upper and lower bound. While the data suggests inhibition, it's too preliminary to be considered for IC50 quantification. This concern is truly for the sake of the rigor of the paper, where more data from additional work is worth the effort.

Round 4

Reviewer 2 Report

The authors attempted to address my concerns but now the abstract peptide sequence shows discrepancies from the peptide sequence in the main text.

The molecular weight is assigned to the second largest peak from mass spec which again raises more concerns . I suggest a thorough check on the peptide identity by rerunning the Edman degradation.

This is because the mass spec mass still differs from the sequence based molecular weight significantly.